# Prospective Evaluation of Neoadjuvant Imatinib Use in Locally Advanced Gastrointestinal Stromal Tumors: Emphasis on the Optimal Duration of Neoadjuvant Imatinib Use, Safety, and Oncological Outcome

**DOI:** 10.3390/cancers11030424

**Published:** 2019-03-25

**Authors:** Shang-Yu Wang, Chiao-En Wu, Chun-Chi Lai, Jen-Shi Chen, Chun-Yi Tsai, Chi-Tung Cheng, Ta-Sen Yeh, Chun-Nan Yeh

**Affiliations:** 1GIST Team, Department of Surgery, Chang Gung Memorial Hospital, Chang Gung University, Taoyuan 333, Taiwan; d0100106@cgu.edu.tw (S.-Y.W.); lairickie@cgmh.org.tw (C.-C.L.); m7202@cgmh.org.tw (C.-Y.T.); atong89130@gmail.com (C.-T.C.); tsy471027@adm.cgmh.org.tw (T.-S.Y.); 2Graduate Institute of Clinical Medical Sciences, Chang Gung University, Taoyuan 333, Taiwan; 3Department of Medical Oncology, Chang Gung Memorial Hospital, Chang Gung University, Taoyuan 333, Taiwan; 8805017@cgmh.org.tw (C.-E.W.); js1101@cgmh.org.tw (J.-S.C.)

**Keywords:** gastrointestinal stromal tumors, GIST, neoadjuvant therapy, imatinib, surgery

## Abstract

Background: Neoadjuvant imatinib therapy has been proposed for routine practice with favorable long-term results for patients with locally advanced gastrointestinal stromal tumors (GISTs). However, clarification of the optimal duration, safety, and oncological outcomes of neoadjuvant imatinib use before surgical intervention remains necessary. Methods: We prospectively analyzed the treatment outcomes of 51 patients with locally advanced, nonmetastatic GISTs treated with neoadjuvant imatinib followed by surgery. The optimal duration was defined as the timepoint when there was a <10% change in the treatment response or a size decrease of less than 5 mm between two consecutive computed tomography scans. Results: Primary tumors were located in the stomach (23/51; 45%), followed by the rectum (17/51; 33%), ileum/jejunum (9/51; 18%), and esophagus (2/51; 4%). The median maximal shrinkage time was 6.1 months, beyond which further treatment may not be beneficial. However, the maximal shrinkage time was 4.3 months for the stomach, 8.6 months for the small bowel and 6.9 months for the rectum. The R0 tumor resection rate in 27 patients after neoadjuvant imatinib and surgery was 81.5%, and 70.4% of resection procedures succeeded in organ preservation. However, 10 of 51 patients (19.6%) had complications following neoadjuvant imatinib use (six from imatinib and four from surgery). Conclusion: Our analysis supports treating GIST patients with neoadjuvant imatinib, which demonstrated favorable long-term results of combined therapy. However, careful monitoring of complications is necessary. The optimal duration of neoadjuvant imatinib use before surgical intervention is, on average, 6.1 months.

## 1. Introduction

Gastrointestinal stromal tumors (GISTs) are the most common mesenchymal tumors of the GI tract and are characteristically driven by activating mutations of KIT in approximately 85–90% of cases [1]. Since 2002, the management of GISTs has been revolutionized with the development of imatinib mesylate, which is a receptor tyrosine kinase inhibitor of KIT, and platelet-derived growth factor receptor-alpha (PDGFRA) [2,3]. Imatinib therapy was initially approved for advanced/metastatic GISTs and subsequently approved for adjuvant therapy after surgery [4]. Although a majority (70–85%) of GISTs are resectable at presentation [5,6,7], the size and/or location of the lesions can make resection challenging, requiring complex operations or leading to permanent lifestyle changes [8].

Prospective and retrospective studies have shown that neoadjuvant imatinib therapy effectively decreases tumor size, thereby facilitating the ease of surgery and resulting in organ-preserving operations with less morbidity [9,10,11]. For example, in patients with duodenal GISTs, a pancreaticoduodenectomy may be converted to local excision of the duodenum, sparing the pancreatic head and common bile duct [12,13]. In patients with rectal GISTs, abdominoperineal resections with permanent end colostomies may be converted to transanal resections, sparing the sphincter [9]. Furthermore, neoadjuvant therapy for GISTs may convert resection from an open laparotomy to a laparoscopic operation.

In the only published multi-institutional trial on neoadjuvant therapy, the therapy was stopped after 8–12 weeks in accordance with the protocol, but a response may occur earlier or well beyond this timepoint [9]. The timing of this plateau response varies between 4 and 12 months [14,15,16]. The National Comprehensive Cancer Network (NCCN) guidelines recommend monitoring the response to neoadjuvant imatinib therapy by imaging until there is no response observed from two consecutive scans or when progression is documented despite escalation of the imatinib dose (plateau response) [4]. However, these recommendations are incomplete, as there is no clear evidence to support this observation and no definite timeframe in this context. Therefore, this study aimed to prospectively determine the timing of plateau responses (as determined by computed tomography (CT) imaging) after the start of neoadjuvant imatinib therapy for GISTs to define the optimal surgical timing, safety, and oncological outcomes for neoadjuvant imatinib use.

## 2. Materials and Methods

### 2.1. Patient Selection and Preoperative Management

From January 2013 to December 2016, we enrolled patients who were diagnosed with locally advanced GISTs without metastasis. For these enrolled subjects, extensive surgeries, including combined esophagectomy and gastrectomy, total gastrectomy, pancreaticoduodenectomy, small bowel resection combined with resection of other organs, and abdominoperineal resection for colonic lesions, were necessary for curative treatment. The treatment plan for each patient was managed by a GIST team consisting of medical oncologists, surgical oncologists, pathologists, and radiologists. Pathological diagnoses were confirmed using standard hematoxylin/eosin staining and CD117 immunohistochemistry on formalin-fixed paraffin-embedded tissues. Patient data were prospectively collected and recorded. This study was approved by the local Institutional Review Board of the Chang Gung Memorial Hospital (101-4844B). Written informed consent for the analysis of tumor-associated genetic alterations was obtained from each patient. This trial was registered at ClinicalTrials.gov in May 2013 (NCT01865565). Clinical data such as demographic data, clinical presentation, response to treatment, surgical condition, postoperative complications, related mutations, and postoperative adjuvant treatment were collected. The last follow-up was performed in December 2017.

### 2.2. Response Assessment

Contrast enhanced CT imaging was performed 1 month after the initiation of neoadjuvant imatinib therapy. Then, follow-up CT imaging was performed every 3 months. We recorded three clinically relevant responses: the earliest response, best response, and plateau response. The earliest response was defined as the earliest time when partial response (PR) was achieved according to the Response Evaluation Criteria In Solid Tumors version 1.1 (RECIST 1.1). The best response was classified according to RECIST 1.1 as a complete response (CR), PR, stable disease (SD), or progressive disease (PD) [17].

The plateau response (or maximal response) was defined as the point when there was <10% treatment response or <5 mm decrease in tumor size between two consecutive CT scans. According to RECIST 1.1, the target lesion was measured by the longest axial diameter instead of the volume. Therefore, we evaluated the response using other criteria for the volume change. The volume criteria were based on a study conducted by Graser et al., where a ≤40% decrease in volume was considered PR and a >33% increase in volume or the appearance of new lesions was considered PD [18,19].

### 2.3. Surgical Indication

Curative surgery needed be conducted when: (1) a plateau in the treatment response was achieved based on the imaging data; (2) a clinically meaningful downstaging in the scope of the planned operation was achieved, beyond which further impact on surgery would be minimal; and (3) a patient became intolerant of the generally limited side effects. Except for the first condition, patients with the other factors were excluded from later analysis. Examples of downstaging of the operation include converting the resection of a gastric GIST from an open laparotomy to a laparoscopic approach and converting the resection of a rectal GIST from an abdominoperineal resection with permanent end colostomy to a sphincter-sparing transanal resection.

Postoperative follow-up consisted of a physical examination and acquisition of contrast-enhanced CT scans at 2- to 3-month intervals or as required by subsequent treatments according to the protocol. The last follow-up was performed in December 2017.

### 2.4. Literature Review

We searched relevant studies related to neoadjuvant imatinib use for locally advanced GISTs. We searched the PubMed database to conduct a primary screening using the following keywords: neoadjuvant, GIST, and imatinib. Subsequently, we chose studies related to neoadjuvant imatinib use in locally advanced GISTs. Both retrospective and prospective studies were included in the literature review.

### 2.5. Statistical Analysis

All statistical analyses and graphing were conducted using R software (version 3.4.3, 2017-11-30, St. Louis, MO, USA) and relevant R packages.

## 3. Results

### 3.1. Demographic Data

From 2013 to 2016, 51 patients treated at the Chang Gung Memorial Hospital, Linkou, Taiwan, were included in this study. The median follow-up period was 46 months. The study subjects included 22 females and 29 males with a median age of 59.9 years. Detailed clinical and pathological data of all patients included in this study are listed in Table 1.

The majority of the primary tumors were located in the stomach (23/51; 55.3%) followed by the rectum (17/51; 20.5%), ileum/jejunum (9/51; 9.3%), and esophagus (2/51; 3.1%). The median time of preoperative imatinib therapy was eight months (range: 16–36 weeks). Six patients (11.8%; 6/51) experienced imatinib-related complications (Table 2); four of the six patients showed hemorrhagic complications, and one patient showed tumor necrosis related to an intra-abdominal infection. 

The sixth patient developed interstitial lung disease, while the other five patients underwent emergency surgery. All six patients were excluded from further analysis. For mutation analysis, we analyzed 37 of the 51 (72.5%) patients in our cohort under the condition of sufficient biopsied tissue availability. The most common pattern of mutation was a solitary exon 11 mutation. The results of the mutation analysis are summarized in Appendix A.

### 3.2. Clinical Response to Imatinib

Figure 1 presents a flowchart of the patient recruitment process. Eleven of the 51 eligible patients were withdrawn from the trial due to disease progression (*n* = 2), surgery for a primary lesion before the maximal response was achieved because of the absence of clinical benefit even with further preoperative use of imatinib (*n* = 3), or treatment complications from imatinib (*n* = 6). Only 40 patients were enrolled for analysis according to the protocol (Figure 1). Among these patients, 38 patients achieved a plateau response. The other two patients remained using medication at the end of the analysis.

Among these 38 patients: 27 patients underwent surgery, and the remaining 11 patients did not undergo surgery for the following reasons: personal choice (*n* = 5), CR (*n* = 1), cancer cachexia due to malignancy other than GIST (*n* = 1), and extremely high anesthesia risk (*n* = 4). The median time required for achieving the earliest PR was 3.7 months. The median best shrinkage percentage in the longest axial diameter was 43% (interquartile range: 31–48%), the volume shrinkage percentage was 83% (interquartile range: 63–87%), and the median time was 6.5 months. The median time for the plateau response was 6.1 months, beyond which further treatment may not be beneficial. The median time for the plateau response was 4.3 months for gastric GISTs, 8.6 months for small bowel tumors, and 6.9 months for rectal tumors (Figure 2).

### 3.3. Surgical Results

The histological status of the margin of resected tumors after preoperative imatinib therapy was R0 in 22 of 27 patients (81.5%) (Table 3). The success rate for organ preservation was 70.4%. For patients with gastric GISTs and failure to preserve adjacent organs, additional procedures included a splenectomy (*n* = 1), a distal pancreatectomy with a splenectomy (*n* = 1), and a cholecystectomy with a duodenectomy (*n* = 1). For patients with small bowel GISTs, a right salpingectomy (*n* = 1) and left hemicolectomy (*n* = 1) were necessary for curative treatment. For rectal lesions, two patients underwent partial vaginal wall resection, and one patient underwent an abdominoperineal resection with a prostatectomy.

Surgical complications were observed in 14.8% (4/27) of the patients and included postoperative ileus (*n* = 2), surgical site hemorrhage (*n* = 1), and acute cholecystitis (*n* = 1). No surgical mortality (death before Postoperative Day 30) was noted. Among all 22 patients who underwent curative surgery, one patient in whom the disease originated in the rectum subsequently experienced local recurrence, while no patient suffered from cancer-related death after curative surgery. The three-year disease-free survival was 95.5% (21/22) after neoadjuvant imatinib with a plateau response and curative surgery (Figure 3). 

### 3.4. Literature Review

Several studies advocate the benefit of neoadjuvant imatinib therapy for locally advanced and/or marginal resectable GISTs [10,20,21,22,23,24]. The relevant studies are summarized in Table 4.

## 4. Discussion

The preoperative use of imatinib appears to be beneficial for patients with locally advanced or marginally resectable primary GISTs. Cytoreduction with imatinib may facilitate R0 resection and organ-sparing surgery [25]. Moreover, because primary tumors are fragile and hypervascular, preoperative imatinib therapy may decrease the risk of bleeding, postoperative complications, and tumor rupture, which is related to a high probability of tumor dissemination [26]. This approach is recommended by the current European Society for Medical Oncology (ESMO) and NCCN guidelines [27,28]. Although several published studies [10,14,20,21,22,23,24,25,29] have advocated the benefit of neoadjuvant imatinib therapy in locally advanced and/or marginally resectable GISTs, most of these studies were small prospective clinical trials or included a small series of patients. Our study aimed to consolidate this practice with a prospective approach and indicated favorable results with this type of clinical practice. In our study, all patients showed either PR or SD after neoadjuvant treatment. The median time required for achieving the earliest PR was 3.7 months. This duration is consistent with the arbitrary duration of neoadjuvant treatment used in the RTOG 0132/ACRIN6665 clinical trial, which was based on the median time for PR in metastatic conditions [9]. Andtbacka et al. found that among 11 patients with locally advanced GISTs treated with neoadjuvant imatinib, nine patients developed a complete response or PR with an absolute median decrease in tumor volume by 85% after a median interval of 48 weeks of imatinib treatment [29]. In a recent study, Tielen et al. reported a median decrease in the tumor size by 50% in 57 patients with locally advanced GISTs who underwent surgery after neoadjuvant imatinib treatment (median duration: 32 weeks; range: 1–55 months) [23]. Consistent with these findings, our study showed that neoadjuvant imatinib therapy decreases the tumor size in locally advanced GISTs.

Regarding surgical intervention after a decrease in tumor size, a multicenter study including 161 patients with locally advanced nonmetastatic GISTs pooled from 10 European Organization for Research and Treatment of Cancer-Soft Tissue and Bone Sarcoma Group (EORTC-STBSG) sarcoma centers showed that >80% of the tumors responded to imatinib, facilitating R0 resection in >80% of the cases [24]. The prospective phase II APOLLON trial that evaluated the use of neoadjuvant imatinib in patients with locally advanced nonmetastatic GISTs (*n* = 41) showed that among 34 patients who underwent surgical resection after a median duration of 27 weeks of neoadjuvant therapy, R0 resection was achieved in 30 patients [10]. All these results imply that preoperative administration of imatinib increases the possibility of complete excision of tumors with a substantial decrease in the need for the removal of surrounding organs. Our prospective trial also supported these results. Our study not only was one of the largest studies but also defined the best timing for surgical intervention, which has not been investigated in previous published studies.

Although the safety of neoadjuvant imatinib therapy for locally advanced disease has not been well addressed, several studies have been published on the use of tyrosine kinase inhibitors for metastatic disease before salvage surgery [30,31,32,33]. In our previous studies on tyrosine kinase inhibitors for metastatic diseases, among the patients administered imatinib and sunitinib, 13.2% (5/38) and 15.3% (4/26) of the patients, respectively, who subsequently underwent salvage surgery had surgical complications [31,32]. Raut et al. also claimed that surgery is feasible for patients with metastatic GISTs who are treated with sunitinib; however, incomplete resections are frequent, and surgical complications occur in 54% of the subjects. Some small prospective series and retrospective studies have proposed high surgical complication rates associated with neoadjuvant imatinib therapy for locally advanced diseases (grade 4 event: >20%) [34]. In the present study, the surgical complication rate was 14.8% (4/27): two patients showed grade 1 complications, and two other patients showed grade 3 complications. The severity and incidence of surgical complications in our study were lower than those in previous studies. In addition, complications related to imatinib should not be ignored. The reasons for the high complication rates observed in our study may be multifactorial. One of our latest studies showed dysregulation of PDGFR and matrix metalloproteinases and an altered microarchitecture of tissues despite withdrawal of tyrosine kinase inhibitors several days before the operation [35]. Although the use of neoadjuvant therapy for locally advanced diseases may increase the risk of complications related to imatinib use before surgery and/or during surgery, meticulous regular evaluation before and during surgery by experienced professionals should lead to favorable outcomes.

The long-term outcome of neoadjuvant imatinib therapy for locally advanced diseases has been reported. In 2013, Tielen et al. summarized several important studies related to neoadjuvant imatinib use [23]. In general, a favorable outcome, i.e., progression-free survival and overall survival, is expected. In the present study, one patient experienced recurrence among the 22 patients who underwent R0 resection. Therefore, neoadjuvant imatinib therapy may not only reduce the extent of required surgical procedures but also provide a favorable outcome. 

There were several limitations of this study. First, it is difficult to directly compare the outcomes of patients undergoing extensive surgery without preoperative imatinib therapy with those of patients undergoing surgery after neoadjuvant imatinib. Although this practice has been endorsed by current clinical guidelines and a better postoperative life quality has been assumed due to the organ-sparing procedure, we were unable to assess the actual survival effect based on our results. Second, we could not obtain sufficient tumor tissue from some participants, especially those who did not undergo surgery after imatinib treatment. This further impeded the mitotic count analysis and mutation analysis. However, the safety of participants, rather than ensuring sufficient biopsy tissue, was our primary concern. Finally, the period of postoperative observation was not long enough in the present study. Long-term follow-up and investigation of the impact of postoperative adjuvant therapies should be considered to develop better treatment plans for locally advanced or marginally resectable primary GISTs. 

## 5. Conclusions

The present study supports the current recommendations of the ESMO and NCCN guidelines for neoadjuvant imatinib use. For locally advanced GISTs, neoadjuvant imatinib therapy shows positive effects by decreasing the tumor size, limiting surgical procedures, and providing favorable long-term outcomes. Meticulous evaluations, regular clinical check-ups, and high-quality surgeries should be performed in both the pre- and postoperative stages for patients with locally advanced GISTs to increase the safety of neoadjuvant treatment.

## Figures and Tables

**Figure 1 cancers-11-00424-f001:**
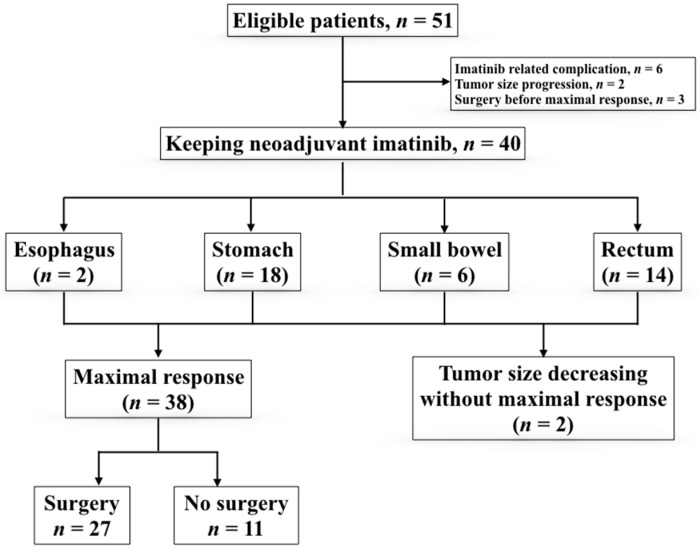
Flowchart of stratification of patients.

**Figure 2 cancers-11-00424-f002:**
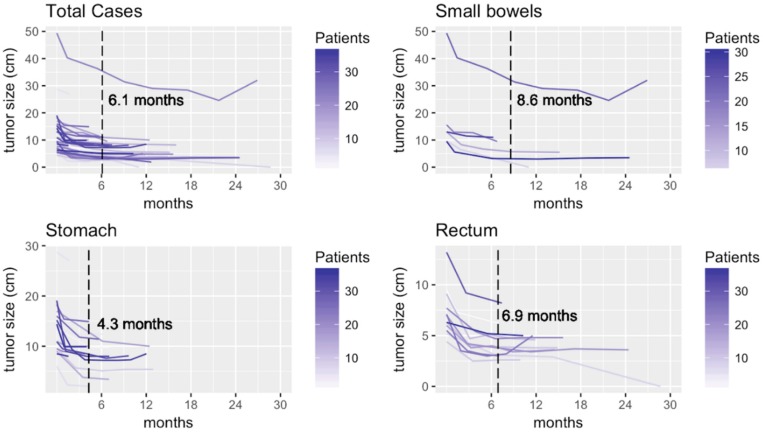
Change of tumor sizes of the per protocol cohort (**upper left**) and different locations of tumors: (**lower left**) for stomach; (**upper right**) for small bowels; and (**lower right**) for rectum.

**Figure 3 cancers-11-00424-f003:**
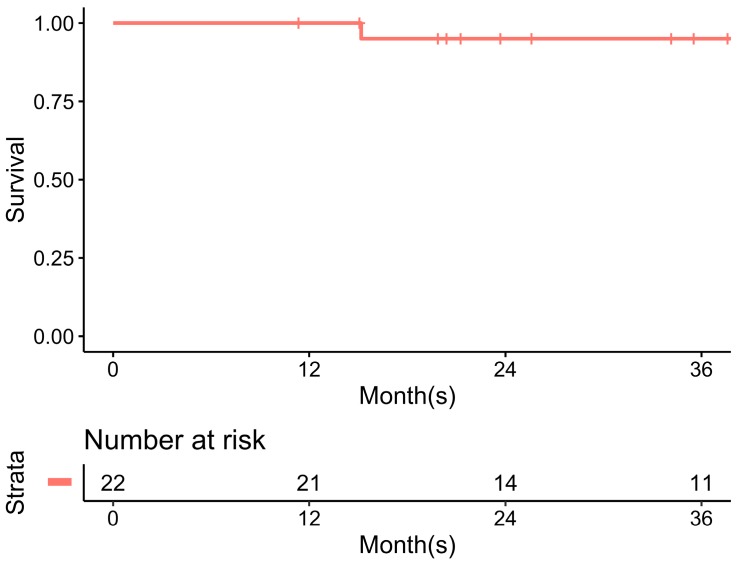
Disease free survival after neoadjuvant imatinib with plateau response and curative surgery.

**Table 1 cancers-11-00424-t001:** Demographic data of 51 eligible patients.

Characteristic	Total Patients (*n* = 51)	Esophagus (*n* = 2)	Stomach (*n* = 23)	Small Bowel (*n* = 9)	Colon/Rectum (*n* = 17)
Age, years (SD)	59.9 (13.1)	73.6 (9.1)	58.3 (12.7)	63.7 (20.2)	58.3 (8.6)
Sex (M/F)	29/22	2/0	13/10	4/5	10/7
Tumor size, cm (SD)	12.5 (7.6)	9.3 (5.3)	14.7 (5.9)	17.5 (12.5)	7.46 (2.33)
WBC, /μL (SD)	6142.1 (2531.4)	8000.0 (141.4)	5873.6 (2629.5)	9220.0 (2631.9)	5407.1 (1484.5)
Platelets, /μL (SD)	214.4K (83.1K)	209.5K (91.2)	207.7K (74.9K)	283.6K (145.8K)	198.7K (56.6K)
Bilirubin, mg/dL (SD)	0.58 (0.32)	0.70 (0.42)	0.62 (0.39)	0.45 (0.06)	0.56 (0.26)
AST, U/L (SD)	26.2 (17.0)	34.0 (32.5)	29.3 (21.2)	21.5 (17.0)	22.5 (5.9)
Cr, mg/dL (SD)	0.92 (0.45)	0.70 (0.42)	0.88 (0.50)	0.97 (0.50)	0.86 (0.23)
AJCC Stage					
Stage I	6	0	5	0	1
Stage II	22	0	11	1	10
Stage IIIa	6	0	1	4	1
Stage IIIb	7	0	3	1	3
Undefined *	10	2	3	3	2
Clinical presentations					
Pain	7	0	5	4	1
Mass	3	0	3	1	1
GI bleeding	11	0	6	1	5
Sign of obstruction	13	2	2	3	7
Incidental finding	5	0	4	0	2
Others	3	0	2	0	1

* Due to inadequate tissue for a mitotic count exam under high-power field inspection, AJCC staging could not be performed for all participants. SD, standard deviation; WBC, white blood cell count; AST, Aspartate transaminase; Cr, creatinine; AJCC, American Joint Committee on Cancer; GI, gastrointestinal.

**Table 2 cancers-11-00424-t002:** Complications after neoadjuvant imatinib use (*n* = 6).

Age/Sex	Time After Imatinib Use (Months)	Tumor Size (Median, cm)	Location	Complications after Imatinib Administration/Postoperative Condition	Resection	Mutation
67M	1.8	19	Stomach	Tumor rupture, intra-abdominal hemorrhage, imatinib stopped after mutation analysis complete/expired within 1 month post-operation	R2	PDGFRA, Exon 18 (D842V)
63M	0.4	18	Stomach	Intra-abdominal hemorrhage, compartment syndrome/expired after 23 months post-operation	R2	KIT, Exon 11
62M	11.5	17.2	Ileum	Enlarged mass with intratumor hemorrhage No recurrence after 28 months post-operation	R0	KIT, Exon 11
64F	0.4	10	Jejunum	Necrosis of tumors, suspicion of abscess formation/no recurrence after 30 months post-operation	R0	KIT, Exon 11
32M	1.8	20	Jejunum	Tumor rupture, intra-abdominal hemorrhage/expired after 2.1 months post-operation	R2	KIT, Exon 11
83F	2.3	10.7	Stomach	Interstitial lung disease	No Surgery	KIT, Exon 11

**Table 3 cancers-11-00424-t003:** Outcome of patients with locally advanced gastrointestinal stromal tumors treated with neoadjuvant imatinib (*n* = 40).

Characteristic		Esophagus	Stomach	Jejunum/Ileum	Rectum	Total (%)
Patients with OP		0	15	2	10	27 (67.5)
Patients without OP		2	3	4	4	13 (32.5)
Time from imatinib use to op (median, months)						8
Resection	R0	0	14	1	7	22 (81.5)
	R1	0	1	0	3	4 (14.8)
	R2	0	0	1	0	1 (3.7)
Organ preservation	Achieved	0	12	0	7	19 (70.4)
	Failed	0	3 ^a^	2 ^b^	3 ^c^	8 (29.6)
Adjuvant imatinib	Yes					24 (88.9)
	No					3 (11.1)
Recurrence		0	0	0	3	3 (11.1)

OP, operation. ^a^ Splenectomy and distal pancreatectomy (*n* = 1), splenectomy (*n* = 1), cholecystectomy and duodenal tumor resection (*n* = 1). ^b^ Appendectomy and right salpingectomy (*n* = 1), En-bloc duodenectomy and resection of jejunum and left hemicolectomy (*n* = 1). ^c^ Partial resection of vagina (*n* = 2), abdominal perineal resection and prostatectomy (*n* = 1).

**Table 4 cancers-11-00424-t004:** Summary of relevant studies on neoadjuvant imatinib therapy.

Reference	Study Type	Patient Number	Conclusions/Statements
Blesius et al. 2011 [20]	Retrospective, part of the BRF14 trial	*n* = 25	Only 9 patients were scheduled for later surgery. Values of overall survival and progression-free survival were close to those for localized intermediate- or high-risk GISTs (70% at 5 years).
Shrikhande et al. 2012 [21]	Retrospective	*n* = 29	Neoadjuvant imatinib therapy for locally advanced GISTs is a safe concept for downsizing, improving resectability, and aiding organ-preserving surgery. It also improves the chance of long-term survival.
Wang et al. 2012 [22]	Prospective	*n* = 53, 31 with primary GISTs and 22 with resectable metastatic/recurrent GISTs	The long-term analysis of the patients enrolled into the RTOG 0132/ACRIN 6665 showed no significant increase in treatment complications after preoperative imatinib use in patients with resectable locally advanced GISTs. A high percentage of patients experienced disease progression after discontinuation of 2-year maintenance imatinib therapy after surgery. Further studies should focus on longer treatment with imatinib.
Hohenberger et al. 2012 [10]	Prospective	*n* = 41	Neoadjuvant treatment with imatinib for 6 months is safe for patients with locally advanced disease. The extent of surgery can be significantly decreased after pretreatment. Even though no adjuvant treatment was foreseen, the postoperative progression-free rate at 3 years is promising.
Tielen et al. 2013 [23]	Retrospective	*n* = 57	Imatinib in locally advanced GISTs is feasible and enables a high complete-resection rate without tumor rupture. The combination of imatinib and surgery in patients with locally advanced GISTs seems to improve overall survival and progression-free survival.
Rutkowski et al. 2013 [24]	Retrospective	*n* = 161	Patients with locally advanced GISTs treated with neoadjuvant imatinib in routine practice show excellent long-term results of combined therapy. Postoperative imatinib therapy should be used routinely in patients considered for neoadjuvant therapy because it is highly unlikely that such tumors are very-low-risk/low-risk GISTs.

GIST, gastrointestinal stromal tumors; RTOG, Radiation Therapy Oncology Group.

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
