# Peer review of "Prospective Evaluation of Neoadjuvant Imatinib Use in Locally Advanced Gastrointestinal Stromal Tumors: Emphasis on the Optimal Duration of Neoadjuvant Imatinib Use, Safety, and Oncological Outcome"

_cancers, 2019, doi:10.3390/cancers11030424_

Round 1
Reviewer 1 Report
Wang S.Y et al present a manuscript entitled “Prospective Evaluation of Neoadjuvant Imatinib Use in Locally Advanced Gastrointestinal Stromal Tumors:
Emphasis on Optimal Duration of Neoadjuvant Imatinib Use, Safety, and Oncological Outcome”
The topic is certainly a hot topic in GIST field. As the authors reported, in 2018 ESMO-Euracan guidelines have been published which are very useful in driving the oncologists in the decision making process.
in my opinion, this paper does not add any novelty to the literature
The ESMO-Euracan guidelines declare that pre-treatment with imatinib is standard but also that “ A biopsy with histological and mutational analyses is recommended to confirm the histological diagnosis, to exclude resistant genotypes to therapy with imatinib (e.g. PDGFRA D842V mutations) and to propose the 800 mg imatinib dose for less sensitive KIT exon 9 mutations.”
Have you considered that? Only 37 patients were analyzed for mutational status. In addition in table 2 a patient D842V is reported. PDGFRA D842V mutation is well described as primary resistant to imatinib
In conclusion, in my opinion, whereas the manuscript is interesting, it is not suitable for apubblication in Cancers
Author Response
Point 1: The topic is certainly a hot topic in GIST field. As the authors reported, in 2018 ESMO-European guidelines have been published which are very useful in driving the oncologists in the decision making process. In my opinion, this paper does not add any novelty to the literature.
Response 1: Neoadjuvant imatinib therapy has been proposed for routine practice for patients with locally advanced GISTs, even in the ESMO guidelines. However, the optimal duration of neoadjuvant therapy has not been fully clarified. In our study, we utilized not only a prospective cohort but also one of the largest cohorts with primary GISTs. In addition to the confirmation of favorable outcomes after neoadjuvant therapy, we investigated the plateau effect of neoadjuvant therapy and proposed the optimal duration of neoadjuvant imatinib use and the best timing of surgical intervention. This information has not been mentioned in the published literature and is the most important contribution of our study. We believe that the results of our study will impact clinical practice.
In addition, we also pointed out that careful monitoring of complications is necessary in the application of neoadjuvant imatinib. This safety issue is important and possibly underestimated. Complications, due to both imatinib and surgery, were investigated in our cohort. We hope that we can share our experience regarding neoadjuvant imatinib with the publication of our present study.
Point 2:Have you considered that? Only 37 patients were analyzed for mutational status. In addition in table 2 a patient D842V is reported. PDGFRA D842V mutation is well described as primary resistant to imatinib.
Response 2:The reason why we only conducted a mutation analysis was that we could not obtain sufficient tissue for analysis. For example, we could not perform mutation analysis for 2 of our patients with esophageal GISTs due to both the technique and safety issues associated with endoscopy. Patient safety is always the first priority, especially when conducting a trial.
In our manuscript, we reported a patient with a D842V mutation. D842V is definitely a hallmark of imatinib resistance. For this patient, we started imatinib therapy after GIST diagnosis. However, the mutational status result was pending, and efficacy was not observed. Complications related to tumor progression occurred early, and he underwent surgery following a short period of imatinib use. We have clarified this point in the revised Table 2.

Reviewer 2 Report
Thank you for allowing me to review.
This is definitely a great effort to do a prospective study in this rare disease. I have a few comments.
1) Would appreciate if you can specify the imaging in line 85. What type of CT? contrast enhanced or non-contrast etc.
2) GIST multidisciplinary team determined the "resectability" of the tumor at the initial visit. It would add value to the manuscript if you can describe what was the criteria used by the surgeon in different GIST sites to determine "resectable" or "unresectable" tumors.
3) Adding AJCC staging will help the readers in other countries.
4) I did not see the limitations of your study in the manuscript. Please add a paragraph describing the limitations of your study.
Thank you.
Author Response
Point 1: Would appreciate if you can specify the imaging in line 85. What type of CT? contrast enhanced or non-contrast etc.
Response 1: The CT that we used for patient evaluation was contrast enhanced. We have specified this point in our revised manuscript.
Point 2:GIST multidisciplinary team determined the "resectability" of the tumor at the initial visit. It would add value to the manuscript if you can describe what was the criteria used by the surgeon in different GIST sites to determine "resectable" or "unresectable" tumors.
Response 2:In our institute, there are several clinical oncology teams, such as teams for colorectal cancer, stomach cancer, and hepatocellular carcinoma. GISTs, however, are tumors that occur in more than one organ, such as gastric cancer in the stomach and HCC in the liver. Therefore, the most important factor in determining tumor resectability is whether metastatic lesions exist. During the routine conferences, radiologists provide all the detailed information on lesions, including the tomographic relationship between lesions and adjacent organs and possible metastatic lesions. Surgeons from different subspecialties (chest surgeon, GI surgeon, or proctologist) then develop the surgical plan.
Under most circumstances, surgeons can perform radical surgery if no metastasis is present. However, for marginally resectable tumors, radical surgery requires extensive resection of visceral organs, which is a major problem that we addressed in our manuscript. Thus, neoadjuvant therapy is quite important in this condition; therefore, we defined the optimal duration of neoadjuvant therapy.
Point 3:Adding AJCC staging will help the readers in other countries.
Response 3:We have added AJCC staging in our revised manuscript.
Point 4:I did not see the limitations of your study in the manuscript. Please add a paragraph describing the limitations of your study.
Response 4:We have added a paragraph to describe the limitations of the study.

Reviewer 3 Report
1) Did the authors stop imatinib treatment before surgery and for how long?
2) Recurrence-free survival and overall survival of patients should be reported.
Author Response
Point 1: Did the authors stop imatinib treatment before surgery and for how long?
Response 1: This is a very difficult question, and we honestly do not have a good answer. We have published an article (Eur J Surg Oncol2019, 45, 153–159) related to the effect of TKI after its cessation. Based on our results from an animal study, the effect of imatinib is present even 1 week following its cessation. For sunitinib, the effect is even longer. Currently, we still have no scientific evidence from human subjects. The timing of imatinib cessation is dependent on individual clinicians.
Point 2:Recurrence-free survival and overall survival of patients should be reported.
Response 2:We have reported both OS and DFS in the revised manuscript.

Round 2
Reviewer 1 Report
The paper is suitable to be published